# Spatial genetic structure across a hybrid zone between European rabbit subspecies

Fernando Alda[1] and Ignacio Doadrio

Dpto. Biodiversidad y Biología Evolutiva, Museo Nacional de Ciencias Naturales (CSIC), Madrid, Spain

[1] Current affiliation: Center for Bioenvironmental Research, Tulane University, New Orleans, LA, USA

## ABSTRACT

The Iberian Peninsula is the only region in the world where the two existing subspecies of the European rabbit (*Oryctolagus cuniculus*) naturally occur and hybridize. In this study we explore the relative roles of historical and contemporary processes in shaping the spatial genetic structure of the rabbit across its native distribution range, and how they differently affect each subspecies and the hybrid zone. For that purpose we obtained multilocus genotypes and mitochondrial DNA data from 771 rabbits across most of the distribution range of the European rabbit in Spain. Based on the nuclear markers we observed a hierarchical genetic structure firstly comprised by two genetic groups, largely congruent with the mitochondrial lineages and subspecies distributions (*O. c. algirus* and *O. c. cuniculus*), which were subsequently subdivided into seven genetic groups. Geographic distance alone emerged as an important factor explaining genetic differentiation across the whole range, without the need to invoke for the effect for geographical barriers. Additionally, the significantly positive spatial correlation up to a distance of only 100 km supported the idea that differentiation at a local level is of greater importance when considering the species overall genetic structure. When looking at the subspecies, northern populations of *O. c. cuniculus* showed more spatial genetic structure and differentiation than *O. c. algirus*. This could be due to local geographic barriers, limited resources, soil type and/or social behavior limiting dispersal. The hybrid zone showed similar genetic structure to the southern populations but a larger introgression from the northern lineage genome. These differences have been attributed to selection against the hybrids rather than to behavioral differences between subspecies. Ultimately, the genetic structure of the rabbit in its native distribution range is the result of an ensemble of factors, from geographical and ecological, to behavioral and molecular, that hierarchically interact through time and space.

Corresponding author
Fernando Alda, alda.fernando@gmail.com

## INTRODUCTION

In most species, populations are genetically structured. This genetic structure may be a consequence of many factors. Foremost among these, geographical factors may lead to vicariant events and divide populations (*Knowles & Carstens, 2007*), or ecological factors may determine habitat suitability across space, and consequently population connectivity

(*Pilot et al., 2006*). Behavioral traits can also shape the population structure of species, such as family groups in primates, or colonies of social insects (*Shoemaker & Ross, 1996*; *Bradley et al., 2002*). Finally, genetic structure will also result from the balance between gene flow, drift, and the time necessary to reach a balance between both forces (*Hutchison & Templeton, 1999*).

The processes leading to population structure can act at different temporal and spatial scales. Temporarily, historical processes such as isolation in glacial refugia and subsequent expansions can leave detectable signals in the current populations (*Avise, 2000*), as well as contemporary dispersal (*Palsboll, 1999*). Spatially, gene flow may vary among individuals within a geographical region, between adjacent regions, or at larger scales, between populations that presumably have little genetic exchange but share a more ancient genetic history. Thus, investigating population relationships and their spatial patterns of genetic variation is useful in order to infer these hierarchical and interacting processes (*Hedrick, 2005*).

The European rabbit (*Oryctolagus cuniculus* Linnaeus 1758) is a species with worldwide biological and economic importance that has long attracted scientific interest (reviewed in *Ferreira, 2012*). This, along with a well-documented history, has allowed the development of multiple studies on the evolutionary history of this lagomorph from a wide range of molecular (*Ferrand, 1989*; *Biju-Duval et al., 1991*; *Branco, Ferrand & Monnerot, 2000*; *Geraldes et al., 2008*; *Carneiro et al., 2012*), temporal (*Hardy et al., 1994*; *Monnerot et al., 1994*) and geographical perspectives (*Webb et al., 1995*; *Fuller, Wilson & Mather, 1997*; *Surridge et al., 1999b*; *Queney et al., 2000*; *Queney et al., 2001*; *Branco et al., 2002*).

In the Iberian Peninsula, two divergent evolutionary lineages occur and contact each other in the middle of their distribution range (*Branco, Ferrand & Monnerot, 2000*; *Branco et al., 2002*). Studies conducted on uniparentally inherited molecular markers support the existence of two highly differentiated groups: so-called mitochondrial lineage A, predominant in the subspecies *O. c. algirus* (Loche 1867) inhabiting the southwest of the Iberian Peninsula, and lineage B, which predominates *in O. c. cuniculus* in the northeast of the Peninsula (*Biju-Duval et al., 1991*). It is proposed that mitochondrial lineages A and B diverged following isolation in two glacial refugia in the southwestern and northeastern extremes of the Iberian Peninsula, likely during the Quaternary paleoclimatic oscillations. After climatic amelioration, they expanded their ranges and came again into contact to form a secondary contact zone where they hybridize along a northwest–southeast axis (*Branco, Ferrand & Monnerot, 2000*; *Branco et al., 2002*; *Geraldes, Rogel-Gaillard & Ferrand, 2005*; *Geraldes, Ferrand & Nachman, 2006*; *Ferrand & Branco, 2007*).

Hybrid zones like this one are usually interpreted as zones where genetically distinct populations meet and interbreed because, despite genetic differences, they have not reached the status of species and are to some extent interfertile. Therefore, they can be considered as intermediate stages in the process of speciation (*Barton & Hewitt, 1989*; *Harrison, 1993*). Hybrid zones may be ephemeral, resulting from the recent meeting and blending of two divergent lineages, or may have arisen from an ancient contact and last indefinitely. In the former case, many individuals in the center of the zone will resemble the

parental forms, leading to high genetic variance and high linkage disequilibrium between loci. In the latter case, the hybrid zone will consists of individuals that are the product of many generations of hybridization, leading to lower genetic variance and lower linkage disequilibrium (*Brelsford & Irwin, 2009*). Therefore, the comparative study of the genetic structure in a hybrid zone and in the parental populations can provide insights into the evolutionary processes that contribute to its origin and maintenance (*Harrison, 1993*).

Hybrid zones are areas of particular interest for evolutionary studies enabling insight into the initial stages of speciation and reproductive isolation, adaptation and selection, and even behavioral processes (*Hewitt, 1988*; *Barton & Gale, 1993*; *Arnold, 1997*; *Futuyma, 1998*). In the European rabbit, extensive studies in the hybrid zone have evidenced highly contrasting degrees of introgression among loci, or even a complete absence of genetic structure (*Branco, Machado & Ferrand, 1999*; *Queney et al., 2001*; *Geraldes, Ferrand & Nachman, 2006*; *Ferrand & Branco, 2007*; *Campos, Storz & Ferrand, 2008*; *Carneiro, Ferrand & Nachman, 2009*; *Carneiro et al., 2010*; *Carneiro et al., 2013*). The variation in the introgression of autosomal and sexual chromosomes has also revealed different selective pressures across genes and its importance in the reproductive isolation between the two rabbit subspecies (*Geraldes, Ferrand & Nachman, 2006*; *Campos, Storz & Ferrand, 2008*; *Carneiro, Ferrand & Nachman, 2009*; *Carneiro et al., 2010*; *Carneiro et al., 2013*).

However, differences in the spatial genetic structure between the parental lineages, either in allopatry or interacting within the hybrid zone, are still largely unknown, because most population genetics studies on the rabbit in the Iberian Peninsula have either examined genetic variation between lineages, or the area where they come into contact (*Monnerot et al., 1994*; *Queney et al., 2001*; *Branco & Ferrand, 2003*; *Ferrand & Branco, 2007*; *Carneiro, Ferrand & Nachman, 2009*; *Carneiro et al., 2010*), but rarely all together or in a comparative context. Furthermore, so far, studies relating the observed genetic structure to behavioral traits or habitat have only been carried out in regions where the rabbit is a non-native species (*Fuller, Wilson & Mather, 1997*; *Surridge et al., 1999a*; *Surridge et al., 1999b*). In this context, the Iberian Peninsula is unique, since it is the only region in the world where the two subspecies are native and co-occur. Therefore, due to its long evolutionary history in this region, it is expected that a complex ensemble of interacting factors affect the genetic structure of the rabbit at different hierarchical scales.

Thus the main objective of this study was to describe and study the relative roles that historical and contemporary processes have in: (1) shaping the spatial genetic structure of the rabbit in Spain, and (2) determining the differences in genetic variability and structure between populations of *O. c. cuniculus*, *O. c. algirus* and the hybrid zone.

# MATERIALS AND METHODS

## Sample collection

Samples of 771 European rabbits were obtained from 30 localities covering most of their range in Spain (Table S1 and Fig. 1). Sampling was performed mainly on hunting estates by licensed hunters during legal hunting seasons, or during management or restocking activities carried out by local administrations. (Since these activities did not involve any

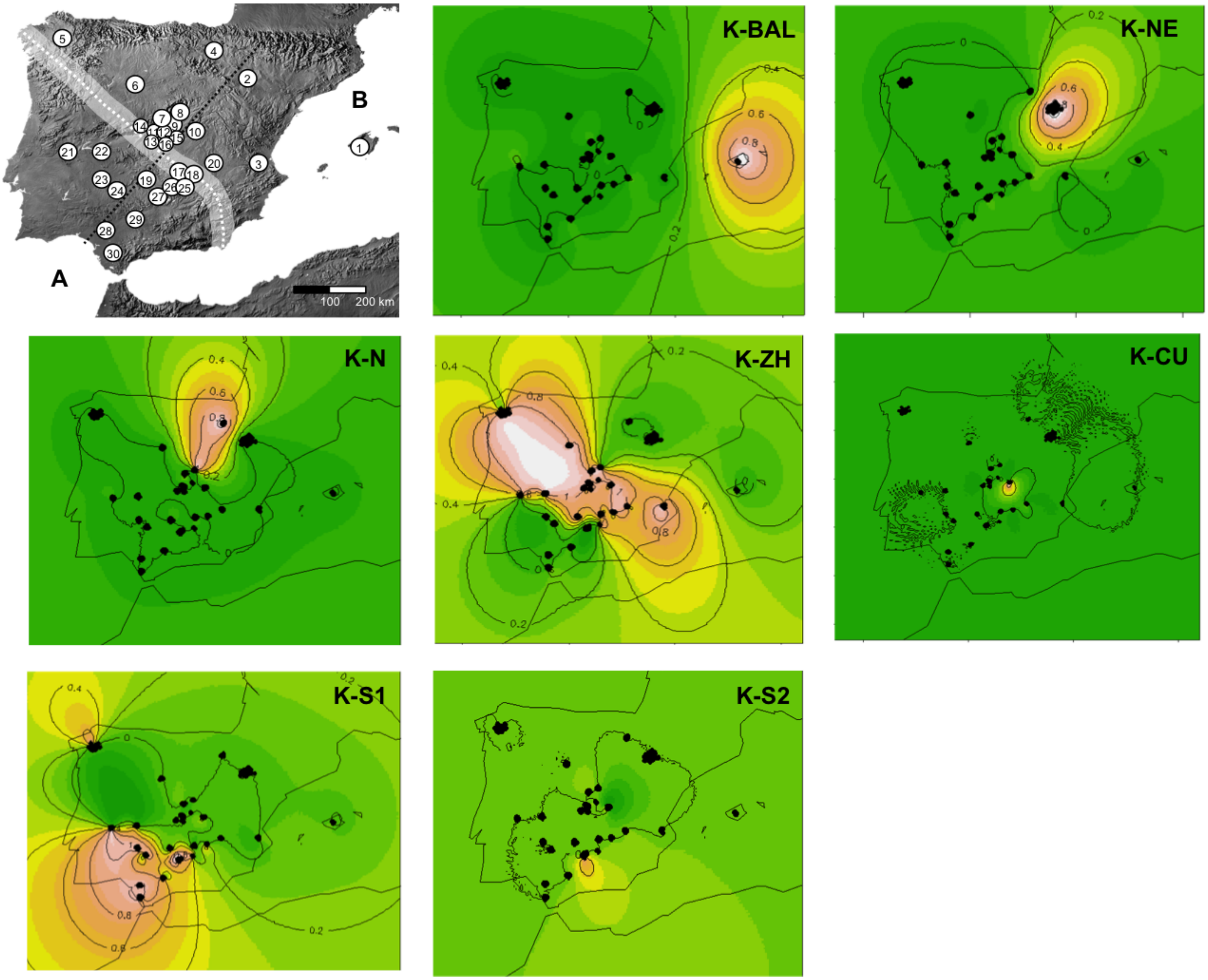

**Figure 1 Maps of the Iberian Peninsula indicating the rabbit localities analyzed.** Maps of the Iberian Peninsula indicating the localities analyzed (numbers correspond to localities in Table S1), the hybrid zone (white dotted line), the perpendicular transect (black dotted line) and the average individual assignment probabilities for the 7 clusters inferred in BAPS. Colour gradient from grey (or green) to white denotes assignment probabilities for each population from 0 to 1.

experimental or scientific purpose, no approval was requested from the Ethics Committee at the Spanish Superior Research Council (CSIC)).

The number of samples per locality ranged from 2 to 56 individuals (Table S1). According to their geographic location (*Branco et al., 2002*; *Geraldes et al., 2008*), 6 localities ($n = 144$) were *a priori* assigned to subspecies *O. c. cuniculus*, 14 localities ($n = 410$) to subspecies *O. c. algirus*, and 10 localities ($n = 217$) to the hybrid zone (Fig. 1, Table S1).

## DNA extraction and amplification of molecular markers

Samples obtained from live rabbits consisted of blood drawn from the femoral vein or a small piece of ear tissue. Ear tissue or muscle samples were taken from dead rabbits, depending on the preservation state of the animal. DNA was extracted using the QIAamp DNA Mini Kit (QIAGEN) following the manufacturer's instructions.

Mitochondrial lineages, A or B, were identified in all samples by amplifying the complete cytochrome *b* gene using the primers OcunCB_F: 5′-ATGACCAACATT CGCAAAACC-3′ and OcunCB_R: 5′-TGTCTCAGGGAGAACTATCTCC-3′. The PCR reaction was performed in a final volume of 25 µL containing: 200 µM of dNTPs, 0.2 µM of each primer, 1 U *Taq* polymerase (Eppendorf), 1× PCR buffer (500 mM KCl, 100 mM Tris-HCl pH8.3, 15 mM $Mg^{2+}$), and 1 µL of DNA extract. The PCR program consisted of 4 min of denaturation at 94 °C, followed by 40 cycles of 1 min at 94 °C, 1 min at 55 °C and 1 min 30 s at 72 °C, plus a final extension of 10 min at 72 °C. PCR products were digested separately with *HaeIII* and *AluI* restriction enzymes (Promega) and migrated in 3% agarose gels stained with ethidium bromide for subsequent visualization under UV light. RFLP patterns were identified as described previously (*Branco, Ferrand & Monnerot, 2000*) and the samples were assigned to one of two mitochondrial lineages.

All individuals were genotyped according to 10 microsatellite markers: Sat3, Sat4, Sat5, Sat7, Sat8, Sat12, Sat13, Sat16, Sol33 and Sol44 (*Mougel, Mounolou & Monnerot, 1997*; *Surridge et al., 1997*). PCR reactions were performed in a final volume of 13 µL that contained 200 µM dNTPs, 0.2–0.4 µM of each primer, 2–2.5 mM $MgCl_2$, 0.325 U *Taq* polymerase (Eppendorf), 1× PCR buffer (500 mM KCl, 100 mM Tris-HCl pH 8.3), and 0.5 µL of DNA extract. PCR programs involved 2 min of initial denaturation at 95 °C, followed by 35 cycles of 30 s at 95 °C, 30 s between 55 °C and 65 °C, 30 s at 72 °C, followed by a final extension step of 7 min at 72 °C. The amplified fragments were analyzed in an ABI3730 automatic sequencer (Applied Biosystems) and allele sizes were assigned using the program GeneMapper v3.7 (Applied Biosystems). The complete data file of microsatellite genotypes and mitochondrial haplotypes was deposited as a Data S1.

## Statistical analyses

Our first objective was to undertake a formal analysis on the genetic structure of the two rabbit lineages and across the hybrid zone between the lineages. For this purpose, we used Bayesian model-based assignment methods to determine admixture proportions in our rabbit sample. Although we were primarily interested in a model with two clusters for assessing admixture between lineages, we also analyzed whether models with more than two clusters were supported by our data. Therefore, firstly we used the algorithm of STRUCTURE 2.2 (*Pritchard, Stephens & Donnelly, 2000*) implementing the admixture model with correlated allele frequencies (*Falush, Stephens & Pritchard, 2003*), since this model is more appropriate for individuals with admixed ancestries and for populations with similar expected frequencies. No information on the localities of origin of individuals was included. Ten independent analyses were run for each value of $K$, from $K = 1$ to $K = 30$. Each analysis consisted of $1 \times 10^6$ Markov chains with a prior burn-in of $1 \times 10^5$

chains. Mean posterior probability values were used to calculate $\Delta K$, a measure of the rate of change of the posterior probabilities between successive $K$ values. Thus, it is possible to detect when the increase in $\ln P(X|D)$ is not significant anymore and find the true value of $K$ (*Evanno, Regnaut & Goudet, 2005*).

To visually explore the distribution of the inferred genetic groups across the hybrid zone, the proportion of genetic admixture and the frequency of each mitochondrial lineage were plotted along a one-dimensional transect perpendicular to the proposed rabbit hybrid zone (*Branco, Ferrand & Monnerot, 2000*; *Branco et al., 2002*; *Geraldes et al., 2008*), with the exception of the localities of Galicia and Mallorca which were excluded (Fig. 1). We considered km 0 of the linear transect to be the approximate intersection between the transect and the hybrid zone (Fig. 1). Geographical locations along this linear transect were fitted to a sigmoid curve (3 parameters) as expected by hybrid zone tension zone models (*Barton & Hewitt, 1985*; *Barton & Gale, 1993*) in SigmaPlot 10.0 (Systat Software, Inc.). Additionally, congruence between the assignment probabilities and the frequency of mitochondrial lineages was evaluated by performing $\chi^2$ tests in both parental lineages and in the hybrid zone.

Secondly, we used the Bayesian method implemented in the program BAPS 5.1 (*Corander & Marttinen, 2006*). In addition to the genetic data, we included the geographical coordinates of each individual and used the spatial model in BAPS (*Corander, Sirén & Arjas, 2008*). This model estimates genetic structure assuming that the structure within a particular area depends on the neighboring areas, thereby increasing the statistical power to detect the true genetic structure (*Corander, Sirén & Arjas, 2008*). We undertook 10 independent replicates from 1 to a maximum of 30 genetic clusters. The average admixture values obtained for each individual were plotted using the *maps* package in *R Core Team (2014)*.

For the inferred genetic clusters and the localities analyzed with more than 10 sampled individuals, we tested significant deviations from Hardy–Weinberg equilibrium through a Fisher exact test (*Guo & Thompson, 1992*) after applying the Bonferroni correction (*Rice, 1989*) in GENEPOP 3.4 (*Raymond & Rousset, 1995*). We calculated parameters of genetic diversity such as number of alleles ($N_A$), allelic richness ($A_R$), observed and expected heterozygosity ($H_o$ and $H_e$) and inbreeding coefficient ($F_{IS}$) for each locus and genetic group using the programs GENETIX 4.02 (*Belkhir et al., 2004*) and FSTAT 2.9.3 (*Goudet, 1995*).

The distribution of genetic variation among the sampled localities, as well as within and among the inferred genetic groups was assessed by an analysis of molecular variance (AMOVA) (*Excoffier, Smouse & Quattro, 1992*). AMOVA was performed using GenoDive 2.0b11 (*Meirmans & Van Tienderen, 2004*) which allows the calculation of a $F_{ST}$ analogue coefficient of differentiation, standardized according to the level of intra-population variation, so that the results obtained can be compared between markers showing different polymorphism (*Meirmans, 2006*). To avoid confusion, this ratio is hereafter referred to as $F_{ST}$'.

The effect of geographical distance on the genetic differentiation between individuals was also tested for all the samples in peninsular Spain (i.e., excluding the population from the island of Mallorca) and for each lineage and the hybrid zone separately. Regression

was performed for the kinship coefficient between pairs of individuals ($f_{ij}$) (*Loiselle et al., 1995*) and their geographical distance ($d_{ij}$), to give a regression slope $\ln b_d$ and its statistical significance (*Vekemans & Hardy, 2004*). Also, spatial autocorrelation methods were applied to examine spatial genetic structure (*Smouse & Peakall, 1999*). Geographic locations of individuals were permuted 10,000 times among 50 distance intervals with an equal number of comparisons between individuals (5,935) to test the null hypothesis that $d_{ij}$ and $f_{ij}$ were not correlated. Positive spatial autocorrelations are expected when gene flow is restricted to short distances. These tests are dependent on the type and scale of sampling (*Vekemans & Hardy, 2004*), so to compare the extent of spatial genetic structure in each subspecies of rabbit and the hybrid zone, we used the statistical *Sp* (*Vekemans & Hardy, 2004*), $Sp = -\ln b_d/(1 - F_{ij})$, where $F_{ij}$ is the average kinship coefficient between individuals closer together (the first distance interval, ∼5 km), and $\ln b_d$ is the slope of the linear regression of the correlation coefficients and the logarithm of the geographical distance. All these tests were conducted in SPAGeDi 1.2 (*Hardy & Vekemans, 2002*). Additionally, a Mantel test (*Mantel, 1967*) was employed to determine if there was significant correlation between the genetic ($F_{ST}/1 - F_{ST}$) and geographical distances of the localities studied. Also, the presence or absence of putative barriers, such as large rivers and/or mountain ranges, between localities was coded as 1 and 0 in a third data matrix. Using these three distance matrices, a partial Mantel test was performed to determine whether, besides geographical distance, these landforms represented a barrier to gene flow for the rabbit. Both analyses were performed in ARLEQUIN 3.1.

# RESULTS

## Distribution of genetic diversity

Overall, similar proportions of rabbits carried mitochondrial haplotypes from lineages A and B (53.5% and 46.5% respectively). In only five localities with over 10 individuals analyzed, all individuals belonged to one lineage. Rabbits from Mallorca, Lérida and Galicia belonged to lineage B, while those from Jaén3 and Sevilla1 belonged to lineage A. Although all the other localities showed a mixture of both lineages, there was a clear predominance of B haplotypes in the northeast of the Iberian Peninsula and Balearic islands, A haplotypes in the southwest, and a mixture of both in the center of the Peninsula (Table 1).

In general, nuclear genetic diversity was high, with a total of 264 alleles at the 10 microsatellites analyzed (average $N_A$ per locus = 26.29 ± 9.07). In all localities heterozygosity values were larger than $H_o = 0.6$ (average $H_o = 0.7 \pm 0.15$). The localities showing greatest diversity, measured as allelic richness, were Madrid1 and Sevilla2 ($A_R = 7.70$ and 7.62), whereas the least diverse were Mallorca ($A_R = 5.15$) and La Rioja ($A_R = 6.20$) (Table 1 and Table S2).

Deviations from Hardy–Weinberg equilibrium were detected in 17 of the 26 locations analyzed (Table 1 and Table S2). In all cases, these deviations were due to heterozygote deficits. Locations in the center and southwest of the Iberian Peninsula showed larger deviations from equilibrium, mainly attributed to locus Sat16 and to a lesser extent

**Table 1** **Genetic diversity statistics based on 10 microsatellite loci genotypes for the rabbit localities (≥ 10 individuals) analyzed and the genetic clusters inferred in BAPS.**

| | $n$ | $N_A$ | $A_R$ | $H_o$ | $H_e$ | $F_{IS}$ | Hap A | Hap B | Subspecies |
|---|---|---|---|---|---|---|---|---|---|
| **Locality** | | | | | | | | | |
| 1. Mallorca | 14 | 6 | 5.15 | 0.63 | 0.71 | 0.15 | 0.08 | 0.92 | *O. c. cuniculus* |
| 2. Lérida | 50 | 9.64 | 6.20 | 0.76 | 0.79 | 0.06 | 0.00 | 1.00 | *O. c. cuniculus* |
| 3. Valencia | 18 | 9.18 | 7.13 | 0.68 | 0.80 | **0.18** | 0.06 | 0.94 | *O. c. cuniculus* |
| 4. La Rioja | 19 | 7.00 | 5.62 | 0.72 | 0.74 | 0.06 | 0.05 | 0.95 | *O. c. cuniculus* |
| 5. Galicia | 27 | 8.73 | 6.42 | 0.61 | 0.80 | **0.26** | 0.00 | 1.00 | *O. c. cuniculus* |
| 6. Valladolid | 16 | 7.91 | 6.41 | 0.67 | 0.73 | 0.11 | 0.07 | 0.93 | *O. c. cuniculus* |
| 7. Madrid1 | 51 | 12.45 | 7.70 | 0.68 | 0.86 | **0.22** | 0.40 | 0.60 | hybrid zone |
| 10. Cuenca | 42 | 12.18 | 7.40 | 0.75 | 0.82 | **0.11** | 0.02 | 0.98 | hybrid zone |
| 11. Toledo1 | 26 | 8.91 | 6.61 | 0.72 | 0.81 | **0.13** | 0.33 | 0.67 | hybrid zone |
| 12. Toledo2 | 33 | 10.91 | 7.09 | 0.68 | 0.83 | **0.19** | 0.40 | 0.60 | hybrid zone |
| 13. Toledo3 | 24 | 10.45 | 7.08 | 0.77 | 0.83 | **0.09** | 0.60 | 0.40 | hybrid zone |
| 15. Toledo5 | 19 | 8.73 | 6.99 | 0.68 | 0.81 | **0.19** | 0.37 | 0.63 | hybrid zone |
| 16. Toledo6 | 11 | 7.82 | 7.08 | 0.78 | 0.82 | 0.09 | 0.36 | 0.64 | hybrid zone |
| 17. Ciudad Real1 | 51 | 12.36 | 7.26 | 0.69 | 0.84 | **0.19** | 0.84 | 0.16 | *O. c. algirus* |
| 18. Ciudad Real2 | 27 | 9.73 | 7.09 | 0.64 | 0.83 | **0.25** | 0.89 | 0.11 | *O. c. algirus* |
| 19. Ciudad Real3 | 50 | 12.27 | 7.34 | 0.70 | 0.82 | **0.16** | 0.82 | 0.18 | *O. c. algirus* |
| 20. Albacete | 25 | 9.64 | 6.86 | 0.75 | 0.81 | 0.10 | 0.48 | 0.52 | *O. c. algirus* |
| 21. Cáceres1 | 10 | 6.64 | 6.22 | 0.70 | 0.72 | **0.10** | 0.90 | 0.10 | *O. c. algirus* |
| 22. Cáceres2 | 28 | 9.00 | 6.51 | 0.63 | 0.78 | 0.21 | 0.89 | 0.11 | *O. c. algirus* |
| 23. Badajoz1 | 20 | 9.73 | 6.82 | 0.71 | 0.79 | 0.13 | 0.95 | 0.05 | *O. c. algirus* |
| 24. Badajoz2 | 29 | 9.45 | 6.99 | 0.79 | 0.82 | 0.05 | 0.90 | 0.10 | *O. c. algirus* |
| 25. Jaén1 | 15 | 8.64 | 6.89 | 0.63 | 0.78 | **0.23** | 0.00 | 1.00 | *O. c. algirus* |
| 27. Jaén3 | 22 | 9.64 | 7.13 | 0.70 | 0.82 | **0.16** | 1.00 | 0.00 | *O. c. algirus* |
| 28. Sevilla1 | 43 | 11.36 | 6.89 | 0.72 | 0.80 | **0.10** | 1.00 | 0.00 | *O. c. algirus* |
| 29. Sevilla2 | 32 | 11.82 | 7.62 | 0.72 | 0.83 | **0.14** | 0.63 | 0.38 | *O. c. algirus* |
| 30. Cádiz | 56 | 11.64 | 6.92 | 0.69 | 0.80 | **0.14** | 0.96 | 0.04 | *O. c. algirus* |
| **Cluster** | | | | | | | | | |
| K-BAL | 14 | 6.00 | 4.35 | 0.63 | 0.71 | 0.15 | 0 | 1.00 | |
| K-NE | 52 | 9.55 | 5.09 | 0.74 | 0.80 | 0.07 | 0 | 1.00 | |
| K-N | 21 | 7.00 | 4.61 | 0.69 | 0.74 | 0.10 | 0 | 1.00 | |
| K-CU | 14 | 6.73 | 4.99 | 0.69 | 0.74 | 0.13 | 0.07 | 0.93 | |
| K-ZH | 457 | 21.00 | 6.20 | 0.70 | 0.87 | **0.20** | 0.49 | 0.52 | |
| K-S1 | 206 | 16.36 | 5.72 | 0.71 | 0.83 | **0.14** | 1.00 | 0.00 | |

**Notes.**

$F_{IS}$ values in bold represent significant deviations from Hardy–Weinberg equilibrium, after Bonferroni correction.
$n$, number of samples; $N_A$, mean number of alleles per locus; $A_R$, allelic richness; $H_o$, observed heterozygosity; $H_e$, expected heterozygosity; $F_{IS}$, inbreeding coefficient.
The proportion of haplotypes from rabbit A and B lineages is shown for each locality and cluster. Numbers of each locality correspond to those in Fig. 1.

to Sat3 and Sol33. Interestingly, none of these loci appeared to be in disequilibrium in the northeastern localities. A comparison of the observed genotypes with a random distribution of genotypes generated by MICRO-CHECKER (*Van Oosterhout et al., 2004*) suggested the presence of null alleles at locus Sat16, as had been proposed earlier for this microsatellite (*Queney et al., 2001*).

## Structure and assignment of rabbit genetic clusters

The Bayesian clustering analysis performed in STRUCTURE revealed that $\ln P(X|D)$ increased substantially from $K = 1$ to $K = 2$ and then was attenuated as the number of $K$ increased but without reaching a clear asymptote. Notwithstanding, calculation of $\Delta K$ clearly revealed the existence of 2 genetic populations or groups (K1 and K2). Taken together, these results suggest that the sampled rabbits belong to two large and distinct genetic groups, but do not completely exclude the possibility that more gene pools exist. The distribution of the two genetic groups exhibited high geographical correlation such that the localities to the south of the Iberian Peninsula were assigned with a greater likelihood to group K1 and the northern localities to K2 group (Table 2 and Figs. 2 and 3).

Both the assignment probabilities to the genetic groups based on the nuclear DNA and the frequencies of mitochondrial lineages closely conformed to sigmoidal functions ($R^2 = 0.878$, $F = 57.478$, $P < 0.0001$ for the microsatellite data; $R^2 = 0.628$, $F = 13.501$, $P < 0.0001$ for the mtDNA data; Fig. 3). In addition, the inferred genetic groups were in agreement with the mitochondrial haplotypes of each individual. Of the 224 individuals assigned to K1 with a posterior probability greater than 0.9, 91% carried mitochondrial haplotypes belonging to lineage A, while 82% of the 239 rabbits assigned with equal probability to K2 showed haplotypes from lineage B (Fig. 2). Thus, the frequency of mitochondrial haplotypes in the parental lineages did not differ significantly from the individuals' assignment frequencies to the inferred genetic groups ($\chi^2 = 0.781$, $df = 1, P = 0.377$).

The analysis in BAPS of the genetic data together with individual geographical information, detected a maximum marginal likelihood (corresponding to the maximum posterior probability) for 7 genetic clusters. Most of the inferred genetic clusters showed a well-defined geographical distribution (Table 2 and Fig. 1). The first cluster corresponded to all individuals from the Balearic Islands (K-BAL). In the north of the Iberian Peninsula, a second cluster appeared consisting mainly of individuals from Lérida (K-NE) and a third cluster comprising individuals from the localities of La Rioja, Madrid1 and Madrid2 (K-N). In the south of the Iberian Peninsula, one large cluster was inferred, that included most of the individuals from Badajoz1, Badajoz2, Jaén2, Sevilla1, Sevilla2 and Cádiz (K-S1), and a small group of individuals from Sevilla2 (K-S2). At the heart of the Iberian Peninsula we found a large cluster fully or partially covering the localities of Galicia, Valladolid, Madrid1-3, Cuenca, Toledo1-6, Albacete, Ciudad Real1-3 and Cáceres1 and Cáceres2 (K-ZH) and another small group of individuals from the locality of Cuenca (K-CU) (Table 2 and Fig. 1). Also, these clusters were congruent with the genetic groups inferred in STRUCTURE and with the mitochondrial haplotypes of the individuals, so

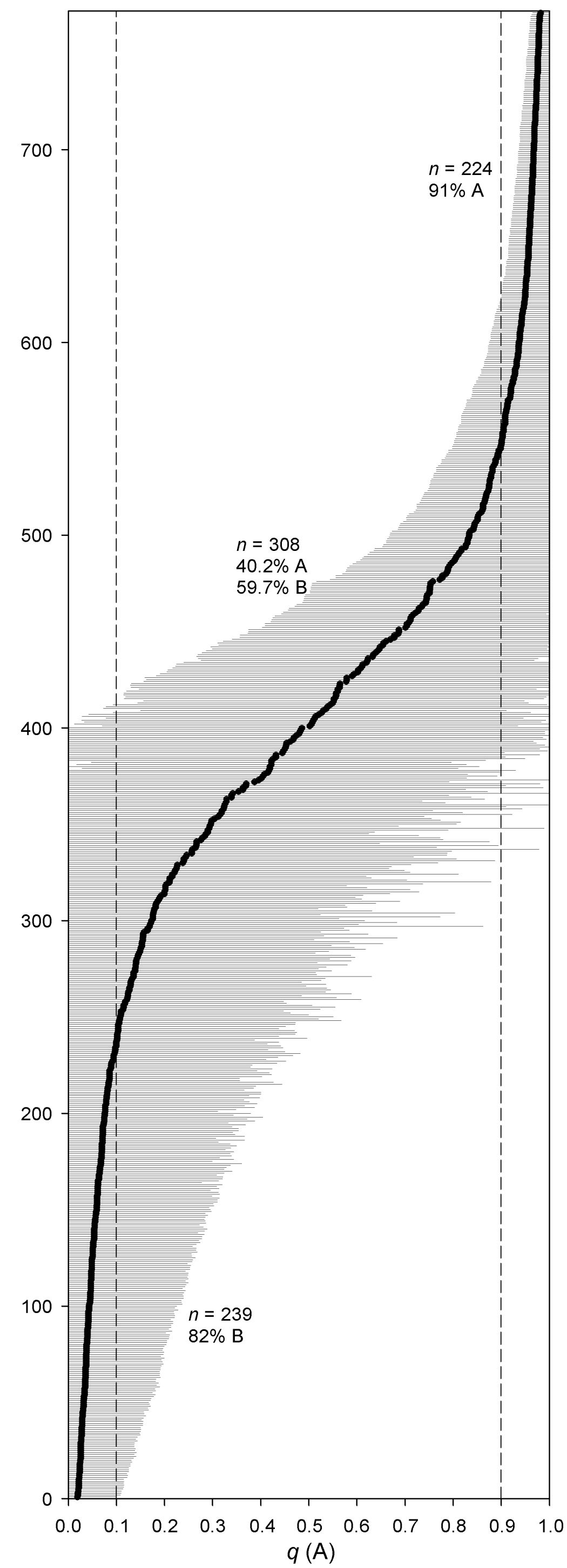

**Figure 2 Individual assignment probabilities to the genetic groups inferred in STRUCTURE.** Individual assignment probabilities ($q$) to genetic group K1 (A). Each dot represents an individual, and grey bars are the confidence intervals obtained for their assignment probabilities. Dashed lines indicate assignment probabilities to group K1 larger than 0.9 and lower than 0.1. The number of individuals assigned within these intervals and the proportion of their mitochondrial lineages are indicated.

**Table 2 Proportion of mitochondrial lineage and average assignment probabilities for each locality to the genetic populations inferred.** Proportion of lineage A and B rabbits and average assignment probability of each locality to the populations inferred in STRUCTURE and BAPS. Numbers of each locality correspond to those in Fig. 1.

| | | mtDNA | | Structure | | BAPS | | | | | | |
|---|---|---|---|---|---|---|---|---|---|---|---|---|
| | *n* | Hap A | Hap B | K1 | K2 | K-BAL | K-NE | K-N | K-CU | K-ZH | K-S1 | K-S2 |
| **Locality** | | | | | | | | | | | | |
| 1. Mallorca | 14 | 0.08 | 0.92 | 0.09 | 0.91 | 1.00 | | | | | | |
| 2. Lérida | 50 | | 1.00 | 0.04 | 0.96 | | 0.98 | | | 0.02 | | |
| 3. Valencia | 18 | 0.06 | 0.94 | 0.26 | 0.74 | | 0.11 | | | 0.89 | | |
| 4. La Rioja | 19 | 0.05 | 0.95 | 0.06 | 0.94 | | | 0.95 | | 0.05 | | |
| 5. Galicia | 27 | | 1.00 | 0.44 | 0.56 | | | | | 0.89 | 0.11 | |
| 6. Valladolid | 16 | 0.07 | 0.93 | 0.19 | 0.82 | | 0.06 | | | 0.88 | 0.06 | |
| 7. Madrid1 | 51 | 0.40 | 0.60 | 0.38 | 0.62 | | | 0.02 | | 0.75 | 0.24 | |
| 8. Madrid2 | 7 | 0.29 | 0.71 | 0.06 | 0.94 | | | 0.29 | | 0.71 | | |
| 9. Madrid3 | 2 | 1.00 | | 0.15 | 0.85 | | | | | 1.00 | | |
| 10. Cuenca | 42 | 0.02 | 0.98 | 0.12 | 0.88 | | | | 0.33 | 0.67 | | |
| 11. Toledo1 | 26 | 0.33 | 0.67 | 0.23 | 0.77 | | | | | 1.00 | | |
| 12. Toledo2 | 33 | 0.40 | 0.60 | 0.19 | 0.81 | | | | | 1.00 | | |
| 13. Toledo3 | 24 | 0.60 | 0.40 | 0.16 | 0.84 | | | | | 1.00 | | |
| 14. Toledo4 | 2 | | 1.00 | 0.08 | 0.92 | | | | | 1.00 | | |
| 15. Toledo5 | 19 | 0.37 | 0.63 | 0.16 | 0.84 | | | | | 1.00 | | |
| 16. Toledo6 | 11 | 0.36 | 0.64 | 0.21 | 0.79 | | | | | 1.00 | | |
| 17. Ciudad Real1 | 51 | 0.84 | 0.16 | 0.67 | 0.33 | | | | | 0.92 | 0.08 | |
| 18. Ciudad Real2 | 27 | 0.89 | 0.11 | 0.47 | 0.53 | | | | | 0.96 | 0.04 | |
| 19. Ciudad Real3 | 50 | 0.82 | 0.18 | 0.66 | 0.35 | | | | | 0.96 | 0.04 | |
| 20. Albacete | 25 | 0.48 | 0.52 | 0.16 | 0.84 | | | | | 1.00 | | |
| 21. Cáceres1 | 10 | 0.90 | 0.10 | 0.70 | 0.30 | | | | | 0.70 | 0.30 | |
| 22. Cáceres2 | 28 | 0.89 | 0.11 | 0.60 | 0.40 | | | | | 0.89 | 0.11 | |
| 23. Badajoz1 | 20 | 0.95 | 0.05 | 0.92 | 0.09 | | | | | 0.05 | 0.95 | |
| 24. Badajoz2 | 29 | 0.90 | 0.10 | 0.94 | 0.06 | | | | | 0.03 | 0.97 | |
| 25. Jaén1 | 15 | | 1.00 | 0.52 | 0.48 | | | | | 0.50 | 0.50 | |
| 26. Jaén2 | 2 | 0.93 | 0.07 | 0.87 | 0.14 | | | | | 0.47 | 0.53 | |
| 27. Jaén3 | 22 | 1.00 | | 0.82 | 0.18 | | | | | 0.82 | 0.18 | |
| 28. Sevilla1 | 43 | 1.00 | | 0.91 | 0.09 | | | | | 0.02 | 0.98 | |
| 29. Sevilla2 | 32 | 0.63 | 0.38 | 0.86 | 0.14 | | | | | 0.03 | 0.75 | 0.22 |
| 30. Cádiz | 56 | 0.96 | 0.04 | 0.91 | 0.09 | | | | | 0.09 | 0.91 | |
| **Population** | | | | | | | | | | | | |
| K-BAL | 14 | | 1.00 | 0.09 | 0.91 | | | | | | | |
| K-NE | 52 | | 1.00 | 0.04 | 0.96 | | | | | | | |
| K-N | 21 | | 1.00 | 0.05 | 0.95 | | | | | | | |
| K-CU | 14 | 0.07 | 0.93 | 0.06 | 0.94 | | | | | | | |
| K-ZH | 457 | 0.49 | 0.52 | 0.37 | 0.63 | | | | | | | |
| K-S1 | 206 | 0.90 | 0.10 | 0.93 | 0.06 | | | | | | | |
| K-S2 | 7 | 1.00 | | 0.89 | 0.11 | | | | | | | |

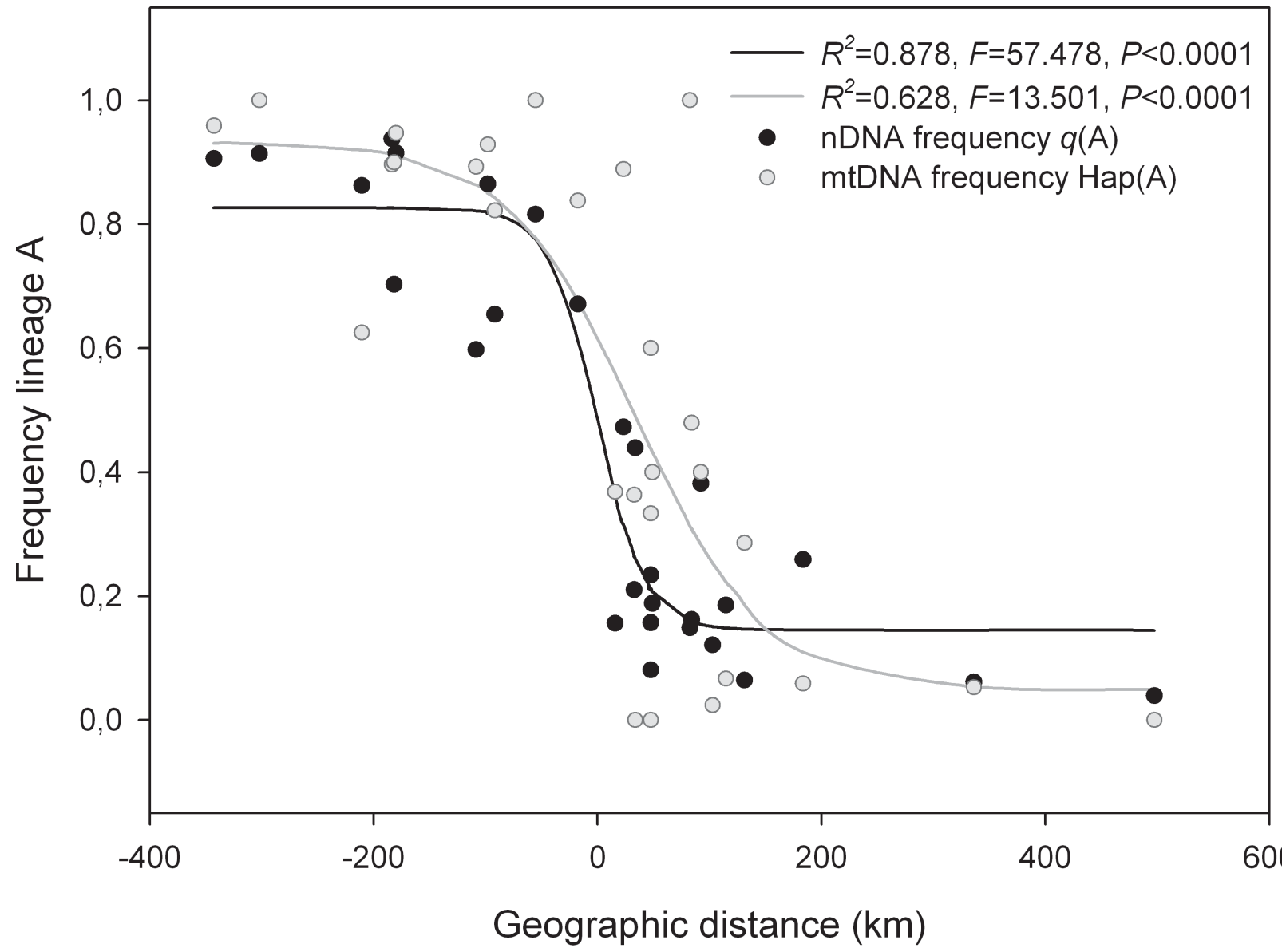

**Figure 3 Clinal patterns for the mitochondrial and nuclear markers along the rabbit hybrid zone.** Clinal patterns for the mitochondrial (grey lines) and nuclear markers (black lines) along the hybrid zone transect from southwestern to northeastern Spain (see Fig. 1). Dots represent the frequency of lineage A mitochondrial haplotypes (grey) and mean assignment probabilities ($q$) to genetic group K1 (A) in each locality. Distances are in km, starting (km 0) at the intersection between the transect and the hybrid zone. Negative distance values indicate km to the south and positive values km to the north.

**Table 3 AMOVA analyses.** AMOVA analyses performed for different levels of genetic structure among the rabbit localities analyzed and the inferred clusters.

| Structure | % Variation | | $F'$ | $p$ | | $R$ | $p$ |
|---|---|---|---|---|---|---|---|
| **Localities** | | | | | | | |
| All localities | 0.287 | $F'_{ST}$ | 0.37 | 0.001 | $R_{ST}$ | 0.11 | 0.001 |
| A Haplotypes Vs. B Haplotypes | 0.111 | $F'_{ST}$ | 0.173 | 0.001 | $R_{ST}$ | 0.637 | 0.001 |
| **Clusters** | | | | | | | |
| STRUCTURE K1 Vs. K2 | 0.192 | $F'_{ST}$ | 0.26 | 0.001 | $R_{ST}$ | 0.635 | 0.001 |
| All clusters BAPS | 0.253 | $F'_{ST}$ | 0.325 | 0.001 | $R_{ST}$ | 0.627 | 0.001 |

that the northern and central clusters (K-BAL, K-NE, K-N and K-CU) had assignment probabilities greater than 90% to K2, and in the same way, the southern clusters (K-S1 and K-S2) to K1.

Genetic diversity parameters estimated for the inferred genetic clusters in BAPS, indicated a greater diversity for K-ZH and K-S1, which were also the only clusters in Hardy–Weinberg disequilibrium, due to a significant deficit of heterozygotes (Table 2 and Table S3). All genetic clusters displayed unique alleles which were usually found at low frequencies. In those genetic clusters that mostly include lineage A rabbits (K-S1 and K-S2), 37 unique alleles were found. While clusters that mostly include lineage B rabbits

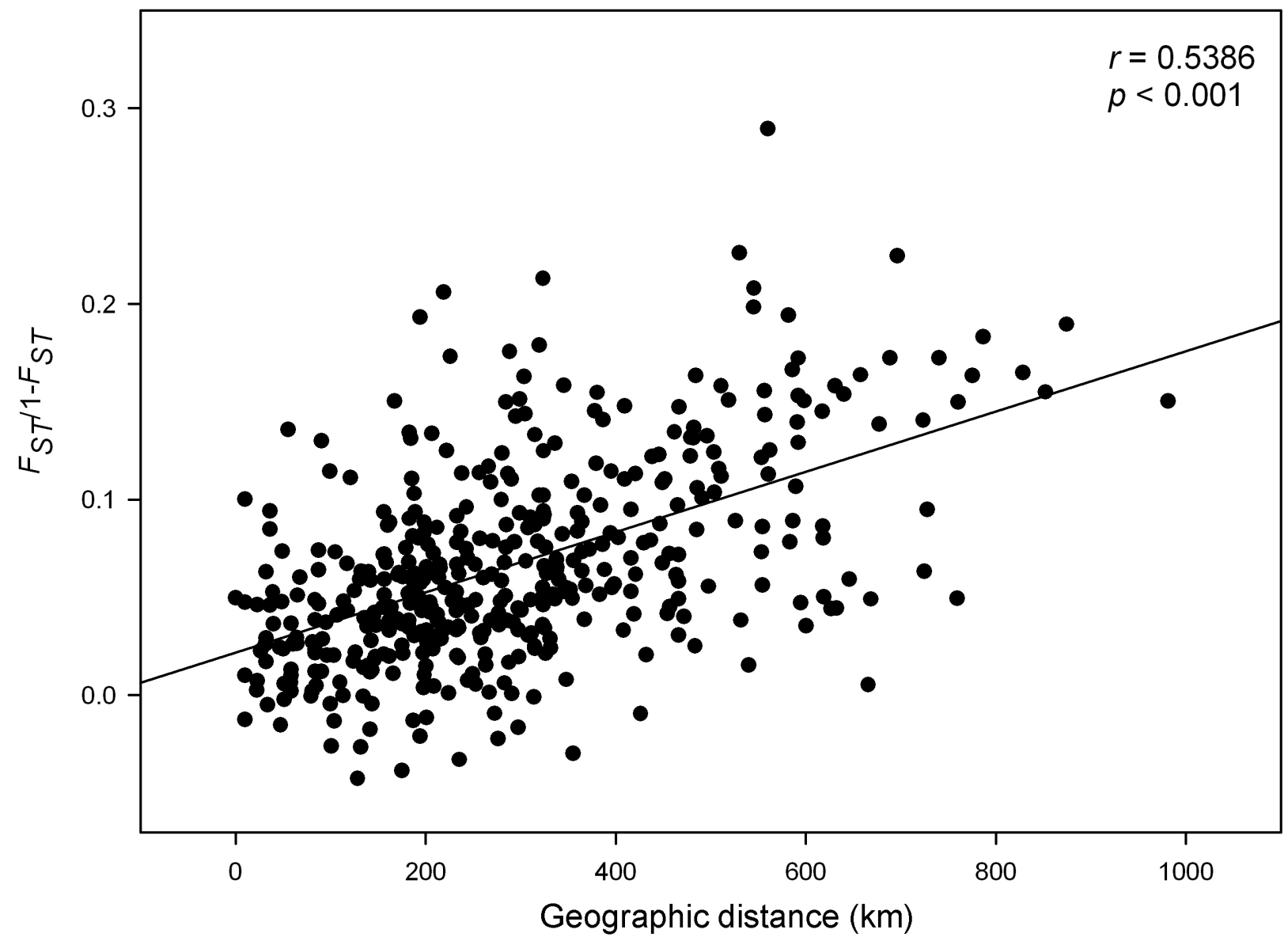

**Figure 4 Isolation by distance among rabbit localities.** Isolation by distance for all the localities of *O. cuniculus* analyzed, as shown by the correlation of genetic distances ($F_{ST}/1-F_{ST}$) and geographic distances (Mantel test).

(K-BAL, K-NE, KN and K-CU) showed 21 unique alleles. Forty-nine unique alleles were detected in cluster K-ZH.

The percentage of genetic variation explained by the 7 genetic clusters was similar, but not greater than that obtained among all localities ($F'_{ST} = 0.325$, $P < 0.001$, $F'_{ST} = 0.370$, $P < 0.001$, respectively). Conversely, $R_{ST}$ was much higher ($R_{ST} = 0.627$, $P < 0.001$ and $R_{ST} = 0.110$, $P < 0.001$, respectively), indicating that the effect of mutation is of greater importance than drift in the differentiation of rabbit genetic clusters (Table 3).

The Mantel test revealed significant correlation between geographical distances and genetic distances for all pairs of populations ($r = 0.538$, $P < 0.001$, Fig. 4). However, this correlation was not improved by including the effect of geographical barriers, such as rivers or mountain ranges, in the partial Mantel test. Across the whole distribution of rabbits in peninsular Spain, the regression slope between kinship and geographical distance was negative and statistically significant ($\ln b_d = -0.011$, $P < 0.001$). Relationships between individuals decreased rapidly as geographical distances increased and this autocorrelation was significantly positive up to a distance of approximately 100 km (Fig. 5A). By comparing the spatial genetic structure of the two subspecies of rabbit and those of the hybrid zone, spatial autocorrelation analyses indicated much higher $f_{ij}$ values and a steeper decline of kinship with distance in *O. c. cuniculus* for the first distance intervals. Similarly, we obtained a higher *Sp* value for *O. c. cuniculus* ($Sp = 0.0137$) than for *O. c. algirus* and the hybrid zone, which showed similar values ($Sp = 0.0062$ and $Sp = 0.0063$, respectively), indicating a greater genetic structure of rabbit populations in the northeast of the Iberian Peninsula (Fig. 5B).

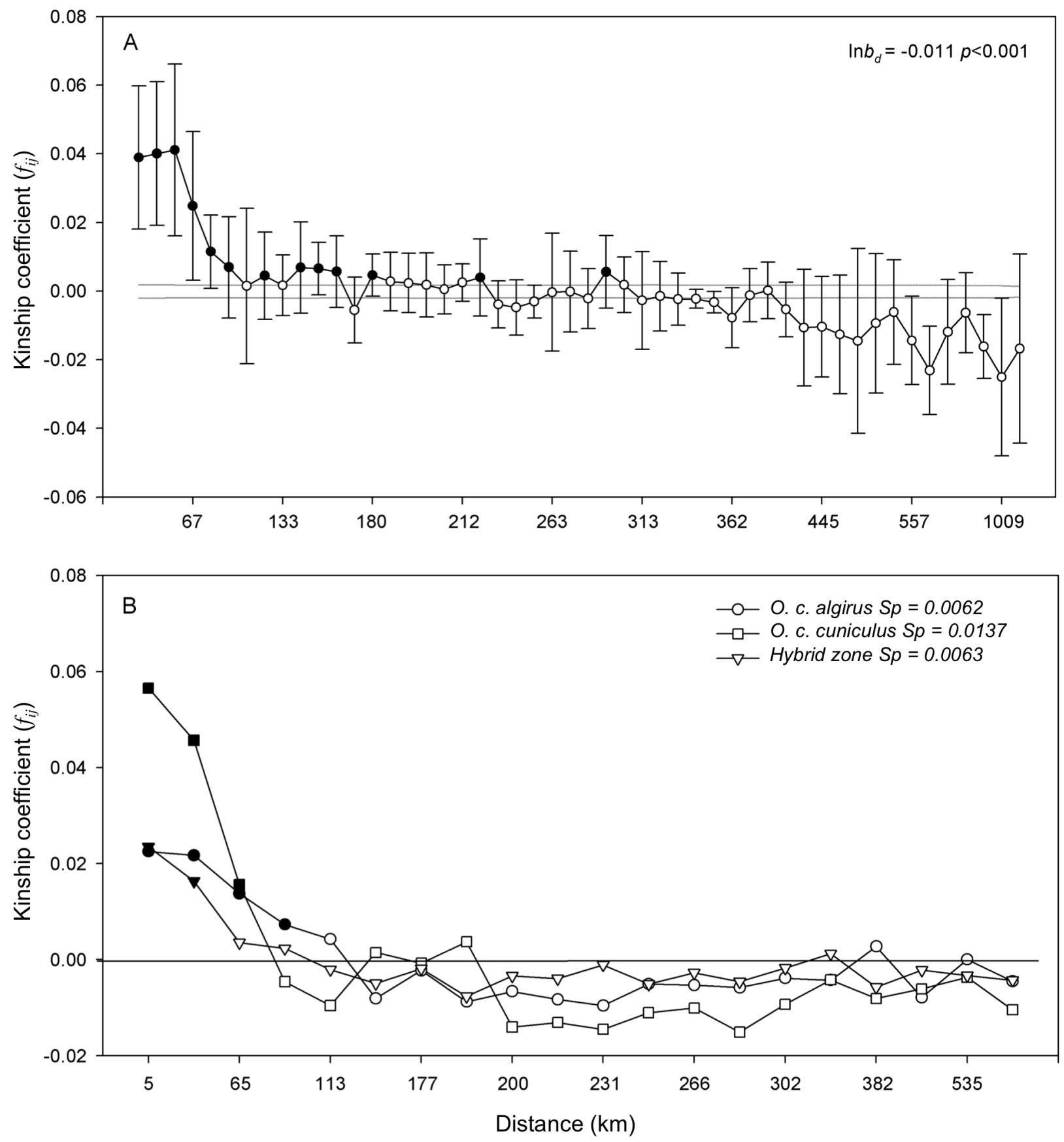

**Figure 5 Spatial autocorrelation analyses.** Spatial autocorrelation analyses showing the average inbreeding coefficient ($f_{ij}$) for each distance interval among individuals ($d_{ij}$), for the complete dataset (A) and for each of the subspecies analyzed and the hybrid zone separately (B). Black symbols represent significant correlations between $f_{ij}$ and $d_{ij}$.

# DISCUSSION

## Variation in genetic diversity

Overall, the microsatellites analyzed were highly polymorphic, and showed similar variability to that reported for 9 of the 10 loci studied (*Queney et al., 2001*). The general trend was greater genetic variability in populations from the central Iberian Peninsula and lower diversity in the northeastern mainland populations (Table 1 and Table S2). This reduced genetic diversity of northeast populations has been attributed to a lower effective size during their isolation in Quaternary glacial refugia, compared to the southern populations (*Branco, Ferrand & Monnerot, 2000*). The lowest diversity and high genetic differentiation found in the island of Mallorca is most likely due to the founder effect caused by the introduction of the rabbit in these Mediterranean islands following the first human arrival to Mallorca 4,300–4,100 years ago (*Flux, 1994*; *Alcover, 2008*). Interestingly, a much older estimate has been proposed for the most recent common ancestor between island and mainland rabbits between 170,000 years and present, according to mitochondrial sequence

data (*Seixas et al., 2014*). While the number of alleles and allelic richness detected for Mallorca were lower than in most of the other samples analyzed, this was not the case for its heterozygosity values (Table 1 and Table S2). Loss of heterozygosity depends on the time it takes a population to recover a large size after a bottleneck (*Nei, Maruyama & Chakraborty, 1975*). Thus, because of its rapid expansion capability, the European rabbit may have managed to retain more diversity during the different colonizations or bottlenecks suffered (*Queney et al., 2000*). Similarly, this could explain why, in La Rioja, where demographic explosions are frequent, a low number of alleles are detected, but not a low heterozygosity. In contrast, in Galicia, both allelic richness and heterozygosity are low and show a significant excess of homozygotes, since rabbit populations in this region have continuously declined in recent years with the consequent loss of genetic diversity (Table 1 and Table S2).

Most localities, particularly the southern ones, revealed loci in Hardy–Weinberg disequilibrium, because of a deficiency in heterozygotes. In this study, as in earlier ones (*Queney et al., 2001*), putative null alleles were detected at Sat16, although the exclusion of this locus and others showing large deviations from equilibrium (e.g., Sat3 and Sol33), did not significantly alter the results. The absence of disequilibrium in the northeastern localities could be explained by the fact that the microsatellites analyzed were originally developed for the domestic rabbit (i.e., subspecies *O. c. cuniculus*) (*Mougel, Mounolou & Monnerot, 1997*; *Surridge et al., 1997*), therefore a higher chance for null alleles could occur in the southern *O. c. algirus*. Another non-exclusive explanation for the significant deficit of heterozygotes could be a Wahlund effect. The territorial behavior and social structure of the rabbit (*Surridge et al., 1999a*; *Surridge et al., 1999b*) could lead to an underlying genetic structure at a small geographical scale that is not detected by Bayesian clustering methods.

## Hierarchical genetic structure dependent on geographic distance

Overall, the European rabbit in Spain showed considerable genetic structure, which was similar to that described for rabbit populations in the northeast and southwest of the Iberian Peninsula (*Queney et al., 2001*) and slightly lower than that reported for Britain (*Surridge et al., 1999a*). The fact that the largest percentage of genetic variation was explained separately by each sampling locality indicates that genetic structure exists at a very local scale and reaffirms the importance of rabbit social behavior in shaping its genetic structure (*Surridge et al., 1999a*).

Bayesian methods and AMOVA, as well as comparisons of $F_{ST}$ and $R_{ST}$ statistics, indicate that the rabbit has a hierarchical genetic structure. First, the oldest and largest differences are mainly reflected by the two genetic groups, based on nuclear markers, and their high $R_{ST}$ values. Within these, there are other genetic groups identified in BAPS that are determined by other factors that could be either environmental or ecological. In turn, these inferred populations consist of even smaller groups conditioned by the social behavior of the rabbit, and are reflected by the significant values of $F_{ST}$ between localities and the significantly positive spatial autocorrelation (*Lugon-Moulin et al., 1999*; *Balloux & Lugon-Moulin, 2002*).

In addition to the hierarchical genetic structure of the rabbit in Spain, geographic distance emerged as an important factor explaining genetic differentiation (Figs. 4 and 5). This contradicts the situation in Britain, where significant differences observed between locations could not be correlated with geographical distance (*Surridge et al., 1999a*). Similarly, it seems logical that main rivers, or other geographical features, constitute a barrier to gene flow in rabbits given their low dispersal capacity (*Webb et al., 1995*; *Richardson et al., 2002*). However, these barriers did not determine an increase in genetic differentiation explained solely by geographic distance. This is probably because, when considering the overall genetic structure and distribution of the rabbit, differences at a more local level have greater importance (*Surridge et al., 1999a*; *Surridge et al., 1999b*; *Branco, Ferrand & Monnerot, 2000*). This hypothesis is further supported by the results of our spatial autocorrelation analyses indicating significantly positive correlation up to a distance of about 100 km (Fig. 5A). However, the influence of geographic distance on genetic differentiation was not the same for all rabbit populations. Northern populations of *O. c. cuniculus* (K2) showed greater relatedness among close individuals and more spatial genetic structure and differentiation than the southern populations or those in the hybrid zone (Fig. 5B). This contrasting pattern could be due to the existence of genetic barriers among populations within each region. For example, it is well known that the Ebro River, running across northeastern Spain, has historically acted both as a physical and an ecological barrier for mammal species (*O'Regan, 2008*). The inferred genetics groups of K-NE and K-N are located to the north and south of the Ebro Valley, thus suggesting its role as a current barrier to gene flow (Fig. 1). Conversely, other large rivers in southern Spain (e.g., Guadiana River) do not seem to hinder gene flow among southern rabbit populations. At a smaller geographic scale, low dispersal might also be due both to resource availability and soil type, which largely influence the distribution and social relationships of rabbits (*Baker & Dunning, 1975*; *Cowan & Garson, 1985*; *Blanco & Villafuerte, 1993*; *Richardson et al., 2002*; *Lombardi et al., 2003*) and of other fossorial mammals (*Lovegrove, 1989*; *Ebensperger & Cofré, 2001*). It has been shown for other burrowing species, such as the wombat (*Lasiorhinus latifrons*), that where soft soils occur the construction of burrows is facilitated so animals do not need to share their shelter with other groups of individuals (*Walker, Taylor & Sunnucks, 2007*). Thus, the social structure of wombats in soft soils is characterized by closely related social groups and positive spatial correlation within a short distance, as observed in *O. c. cuniculus* in the northern Iberian Peninsula where softer soils also exist (*Blanco & Villafuerte, 1993*). In contrast, in hard soils, wombats share burrows with other individuals, and therefore are less related and spatial correlation is observed at a greater distance (*Walker, Taylor & Sunnucks, 2007*), as observed for the southern *O. c. algirus* populations (Fig. 5B).

## Genetic variation within the hybrid zone

The large differences between the two rabbit lineages were evidenced by the maximum $R_{ST}$ value obtained when considering the genetic variation among lineages A and B, which represent a divergence of 1,800,000–2,000,000 years (*Branco, Ferrand & Monnerot, 2000*;

*Carneiro, Ferrand & Nachman, 2009*). The transition between these two genetic groups and mitochondrial lineages is well explained by a sigmoid curve. This was consistent with the Bayesian clustering of STRUCTURE, which indicates that the hybrid zone is not formed by individuals with a bimodal distribution of genotypes from the parental lineages, but instead they form a gradual cline of assignment probabilities to each group (Figs. 2 and 3). On the other hand, when geographic information was incorporated in the Bayesian clustering analysis of BAPS, which usually helps to increase the power of analysis in cases where hierarchical structure might hinder the delineation of discrete groups on a smaller scale (*Corander, Sirén & Arjas, 2008*), the hybrid zone was shown as a large genetic cluster itself (Fig. 1). However, this result should be taken cautiously, since it could represent an artifact of the method. Firstly, Bayesian clustering methods can overestimate genetic structure when analyzing scenarios under a pattern of isolation by distance (*Frantz et al., 2009*), or under strong linkage disequilibrium or departures from Hardy–Weinberg equilibrium (*Falush, Stephens & Pritchard, 2003*). Secondly, a kind of mixture linkage disequilibrium can occur even between physically unlinked loci, due to the correlation of allelic frequencies within populations. As a consequence, highly contrasting parental genotypes can lead to differences in this pattern of linkage disequilibrium and intermediate allele frequencies between these populations be interpreted as a distinct genetic cluster (*Falush, Stephens & Pritchard, 2003*; *Kaeuffer et al., 2007*).

Notwithstanding, hybrid zones can also be characterized by new genotypic combinations, resulting from the crossing of genetically divergent individuals (*Arnold et al., 1999*). As expected from a region comprising the gene pool from both lineages of rabbit, the genetic diversity found in the hybrid zone was higher than in the parental populations, as reflected by the total number of alleles, allelic richness and expected heterozygosity (Table 1). Interestingly, the higher number of alleles was mainly due to 49 alleles exclusively observed in this region, as opposed to the 37 and 21 exclusive alleles found in the parental populations. Though this could simply be the consequence of the higher number of individuals found in this inferred cluster ($n = 457$), it is surprising that the hybrid zone shows so many exclusive alleles, when we would initially expect it to only hold the sum of the parental alleles. Unique alleles have been previously described in the rabbit hybrid zone for the HBA haemoglobin alpha chain gene, which probably originated by recombination of alleles from the parental lineages (*Campos, Storz & Ferrand, 2008*). However, in the case of microsatellite loci, further evidence could suggest that these new alleles might be the result of an increased mutation rate caused by higher heterozygosity of the hybrids (*Bradley et al., 1993*; *Hoffman & Brown, 1995*; *Amos & Harwood, 1998*).

In the hybrid zone cluster, considered as the region with intermediate frequencies not belonging to any of the parental groups, the genetic contribution of each rabbit lineage was not balanced. In this area, the frequency of the two mitochondrial lineages is virtually the same (A = 0.485 and B = 0.515), but significantly greater proportions of individuals had been assigned to K2 (K1 = 0.37 and K2 = 0.63) ($\chi^2 = 25.187$, $df = 1$, $P < 0.0001$), showing a greater genetic introgression of lineage B, characteristic of the northern *O. c. cuniculus*, into lineage A, *O. c. algirus*, than vice versa. This is consistent with recent findings related

to autosomal loci (*Carneiro, Ferrand & Nachman, 2009*; *Carneiro et al., 2013*), yet contrasts with that described for the X chromosome, suggesting slight introgression from the southwest to the northeastern lineages of the Iberian Peninsula (*Geraldes, Ferrand & Nachman, 2006*). The fact that the greatest contribution of the northern rabbit lineage is only reflected in the frequencies of nuclear markers and not in those of maternal inheritance could suggest that males are primarily responsible for this bias. If this is the case, it would be expected that the Y chromosome would be more introgressed than autosomal loci. Conversely, it has been evidenced that the Y chromosome cline is highly stepped, as well as the mtDNA, which suggests some kind of selection acting against introgression (*Geraldes, Ferrand & Nachman, 2006*; *Geraldes et al., 2008*; *Carneiro et al., 2013*). In this regard, preliminary behavioral work discarded the existence of pre-mating reproductive selection between lineages, and found instead lower fertility in F1 males, thus following the expectations of Haldane's rule (*Haldane, 1922*; *Blanco-Aguiar et al., 2010*). In this context it seems that the relative role of selection leading to postzygotic barriers has a stronger importance in shaping the genetic structure in the rabbit hybrid zone than behavioral and prezygotic barriers. Similarly, different types of selection have been detected at several autosomal loci, suggesting a wide range of evolutionary pressures across the rabbit's genome as well as across distribution range in the Iberian Peninsula (*Campos, Storz & Ferrand, 2008*; *Carneiro et al., 2012*; *Carneiro et al., 2013*).

Ultimately, multiple factors ranging from geographical and ecological, to behavioral and molecular, are interacting and shaping the overall genetic structure of the rabbit subspecies and their hybrid zone. Future studies using genomic data coupled with behavioral and ecological information could further clarify how these issues are related to the differences in genetic variation and structure of the rabbit subspecies.

## ACKNOWLEDGEMENTS

The authors wish to thank all those who provided samples for this study: the Consellería de Territori i Habitatge from Generalitat Valenciana, the Hunting Federation of Lérida, Hunting Society of Ajalvir (Madrid), M Sanmartín from the University of Santiago de Compostela, S Agudín, J Inogés, J Layna, F Leiva, F Silvestre and especially to F Guil from Fundación CBD-Habitat. We also thank N Ferrand and S Lopes for data reviewing and helpful discussion of the results, as well as two anonymous referees for their thorough revisions. A Peterson reviewed the English text, L Alcaraz assisted in the laboratory and A Benítez-López provided assistance with statistical analyses.

### Funding

F Alda benefited from a FPU pre-doctoral grant from the Spanish Ministry of Education and Science. Funding was provided by projects MAM/2484/2002-65/2002 and 010203030003 granted to I Doadrio and R Zardoya. The funders had no role in study design, data collection and analysis, decision to publish, or preparation of the manuscript.

## Grant Disclosures

The following grant information was disclosed by the authors:
Spanish Ministry of Education and Science.

## Competing Interests

The authors declare there are no competing interests.

## Author Contributions

- Fernando Alda conceived and designed the experiments, performed the experiments, analyzed the data, contributed reagents/materials/analysis tools, wrote the paper, prepared figures and/or tables, reviewed drafts of the paper.
- Ignacio Doadrio conceived and designed the experiments, contributed reagents/materials/analysis tools, reviewed drafts of the paper.

## Animal Ethics

The following information was supplied relating to ethical approvals (i.e., approving body and any reference numbers):

The study presented did not involve any experiments with live animals. All the animals used in this study were not captured for scientific purposes, but for management and restocking activities carried out by local administrations, or represented animals killed by hunters during legal hunting season in Spain. Therefore, no approval was requested from the Ethics Committee at the Spanish Superior Research Council (CSIC).

## Supplemental Information

Supplemental information for this article can be found online at http://dx.doi.org/10.7717/peerj.582#supplemental-information.

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
