# Peer review of "Spatial genetic structure across a hybrid zone between European rabbit subspecies"

_PeerJ, doi:10.7717/peerj.582_

## Round 0.1 · original submission · Major Revisions

I have received two very thorough reviews, and both reviewers recommend major revision. Unfortunately I cannot recommend acceptance at this time.

I encourage you to read the reviews carefully and respond to the individual comments. Your paper will be sent out a second time for review once you have submitted a revision.

Reviewer 1 ·

Basic reporting

The manuscript by Alda et al. describes analyses of microsatellite variation in ~800 rabbits sampled from multiple localities in the Iberian Peninsula and encompassing most of native distribution range of the species in this region. The authors employ mostly clustering methods to infer patterns of genetic structure and the dynamics of secondary contact. This paper uses a substantial dataset; however, given that the rabbit hybrid zone is now well documented using even larger datasets of both markers and individuals, I am not sure that the paper presents new findings of general interest. Although the use of fast evolving markers such as microsatellites could result in an advance (but not a substantial one), this paper in its current form is mainly about patterns of genetic structure but most elements were already known. I have also some significant concerns with many aspects of the paper and possible over-interpretation in some places, and I detailed them below in no particular order.

Experimental design

1. As mentioned in the basic reporting section, I would like the authors to more clearly highlight what is new here. It seems there is a large overlap with multiple studies of the rabbit hybrid zone already published and the objectives of this study are not clearly stated. The authors attempt this at the last paragraph of the Introduction section but all the items are quite vague. The Discussion section also does not flow well from the Introduction and seems quite rambling in many places. A more focused discussion is needed. The limitations of the experimental design and inference approaches are also not clearly stated throughout the paper.

2. Is there any specific reason why the authors did not attempt cline analysis using alleles frequencies at the 10 microsatellite loci and mtDNA? (see for example Gay et al. 2008; Evolution). Such analysis could provide a more formal framework to test the author’s hypotheses.

Validity of the findings

1. One of my major concerns with the paper is related to the claim that extensive hybridization has generated a genetically differentiated population from the parental ones. As the authors correctly point out in the Introduction section, older hybrid zones between incomplete isolated taxa will consist mainly of individuals that are the product of many generations of hybridization, which will result in introgression across the hybrid zone that may vary across loci. Allele frequency change using SNP data for putatively neutral loci in the rabbit hybrid zone has been reported to be gradual (i.e. no genetic discontinuities are detected) and clinal variation is still observed even for loci showing large cline widths, resembling isolation by distance models (Carneiro et al. 2013). This is in fact also supported by the STRUCTURE assignment probabilities presented in Figure 2 by the authors. My concern is that clustering methods are forcing discontinuities in a system where allele frequencies change gradually across the sampling area. The clustering of individuals in the hybrid zone may simply reflect their intermediate allele frequencies between the parental genotypes, bounded by a step in allele frequencies from one parental type to the other as typically observed in tension zones, such as the rabbit hybrid zone. The fact that genetic structure is inferred within the parental forms (mostly O. c. cuniculus) but all individuals sampled close to the hybrid zone form a single cluster, even when located in opposite ends of the Iberian Peninsula, is consistent with such a scenario.

2. Another major concern is related to the inference and interpretation of asymmetric gene flow across the hybrid zone based on the discordance between the nuclear loci (i.e microsatellites and using assignment probabilities derived from STRUCTURE) and the mtDNA. The authors further conclude that introgression is likely driven by males and invoke pre-mating selection and/or behavioral differences. I think that the logic behind this interpretation is flawed for at least two reasons. First, the authors do not provide details on how they define the hybrid zone localities (subspecies affiliation in Table 1), but if the mtDNA was used this is in my view highly circular. Genetic drift acts in hybrid zones and can underlie variation among loci in patterns of introgression. Given that the mtDNA is a single marker and is therefore subject to stochasticity, if anything, the nuclear genome is likely to provide a better estimate of the hybrid zone location and could easily be invoked that mtDNA is shifted by either demographic or selective forces. Additionally, Carneiro et al (2013) using a larger sample size and more formal cline analysis did not detect statistically different center locations for the mtDNA versus the consensus center using many autosomal and X-linked loci. Second, and unlike in many hybrid zones where mtDNA introgresses pervasively, the mtDNA cline in the rabbit hybrid zone is strongly stepped and characterized by a narrow cline width compatible with selection acting to restrict introgression, thus suggesting that mtDNA may underlie hybrid unfitness. The same is true for the Y that shows an even narrower cline width. Loci on the autosomes and X show a wide range of cline widths and some loci show opposite directionality in their introgression patterns. These contrasting patterns of introgression and asymmetry among loci seem to suggest that selection against hybrid genotypes acting differentially among loci are likely responsible for the observed pattern. For example, in a strict neutral scenario and assuming that dispersal is comparatively more mediated by males than females, as it seems to be the case in most mammals including rabbits, the Y chromosome would be expected to introgress more and that is not the case. I think that a substantially larger dataset and avoiding uniparental markers would be necessary to infer directionally of introgression in the rabbit hybrid zone. The authors should at least acknowledge these issues.

3. Discussion section (Page 18 Line 18). “…one could deduce that sufficient time has passed to reach a balance in which hybrids are not selected against and can have their own genotypic combinations…” The authors should remove this sentence. The shape of clines in the rabbit hybrid zone is consistent with selection acting and, at least to my knowledge, hybrid males suffer from reduced fertility. Many other untested phenotypes are likely to play a role too.

4. In Page 19 Line 407 in the Discussion section it is suggested that new alleles are likely emerging in the rabbit hybrid zone. The authors base this conclusion on two main arguments. First, higher genetic diversity is found in the hybrid zone. Second, the increased number of exclusive alleles also detected in the hybrid zone. The first cannot be used as an argument because increased variation is expected in regions of contact between different taxa, since migration brings together genetic diversity from two structured populations therefore inflating estimates. The second can be used but the authors should test this in a more formal way. It is reported that 49 exclusive alleles are found in the hybrid zone localities but how does this compare to the localities within the territory of both subspecies? This is relevant because microsatellites are fast evolving markers and many more individuals are included in the hybrid zone cluster (457) when compared to the remaining clusters (~200 or much less) so it is not clear whether this is simply a byproduct of unequal sample sizes.


5. The authors report in several places the discordance between divergence times obtained for different genetic markers. Although not great significance should be attached to the actual values due to uncertainties in mutation rate, gene genealogies of the Y-chromosome (Geraldes et al. 2005; 2008) and mtDNA (Branco et al. 2000), as well as autosomal data analyzed under an isolation-with-migration framework (Carneiro et al. 2009), are all consistent in suggesting a divergence time of > 1MYA. Thus, the reported discordance is restricted to allozyme data in which genetic distance based on allele frequencies and a model of no gene flow was used, which clearly does not fit well with the demographic history of rabbit subspecies where introgression seems to have been ubiquitous in their recent evolutionary history. This particular estimate is likely to be an underestimate and should stop being reported.


6. Page16Line343 – The authors go to great lengths in the Discussion section to explain the deviations to Hardy-Weinberg equilibrium inferred in O. c. algirus localities and invoke quite complex explanations. Could the fact that the microsatellites used in this study were ascertained and tested in wild and domestic rabbits belonging to the O. c. cuniculus subspecies (Mougel et al. 1997; Surridge et al. 1997) explain the stronger heterozygote deficiency inferred in O. c. algirus due to a higher incidence of null alleles? Could this ascertainment bias influence some of the analysis performed?


7. The same is true for the explanations associated with the increased substructure inferred in O. c. cuniculus territory when compared to O. c. algirus. This northeastern region of Iberian Peninsula contains large mountains ranges that could act as barriers reducing gene flow between populations. Connecting this increased substructure to food availability and soil type as suggested by the authors, which are more likely to patterning genetic structure on much smaller scales, is in my opinion too speculative.


8. Linkage disequilibrium analyses could perhaps be removed if they only serve the purpose of testing for marker independence, since I believe that this has been done in previous publications. The authors could instead report the genome location of these microsatellites by blasting the primers against the rabbit reference genome sequence.


9. This is up to the authors, but the Mallorca population could perhaps be removed from the data analysis since it is tangential to the main objectives of the manuscript.


10. Page5Line89 – “Similarly the hybrid zone between lineages is more pronounced or varies in size depending on the markers in question.” This sentence should be rephrased. The overall width of a hybrid zone is not a property of genetic markers but of the organisms in question, what changes is the degree and directionality of introgression across the genome.

Additional comments

No Comments

Reviewer 2 ·

Basic reporting

The following represents lack of clarity or poor English:
Line 22: replace ‘in the penetration of a lineage’s genome into the other’ by ‘in the degree of introgression of the different genomes involved in the hybrid zone’
Line 23: replace ‘the two European rabbit subspecies’ by ‘the two subspecies of European rabbit’
Line 32: replace ‘a higher genetic structure’ by ‘more genetic structure’
Line 33-35: replace ‘significantly different frequencies of nuclear genetic groups and mitochondrial lineages in the hybrid zone evidenced’ by ‘significant differences in nuclear and mitochondrial genetic architecture in the hybrid zone indicated’
Line 43: replace ‘yet’ by ‘and’
Line 54-55: replace ‘their agreement can be compared to analyse sex biased contributions, as well as differences in the penetration of a species’ or lineage’s genome into the other’ by ‘it is possible to analyse sex biased contributions and other types of differential introgression’.
Line 57: replace ’designed to gain’ by ‘enabling’
Line 62: add word: ‘as in the European rabbit’
Lines 91-95: sentence not clear – needs rewriting.
Line 105: replace ‘this region’ by ‘the zone’.
Line 146: after MgCl it should be a subscript ‘2’
Line 238: is this average NA per locus?
Line 238: replace ‘rendered’ by ‘generated’
Line 376: replace ‘evidenced’ by ‘shown’
Table 1: more information is needed. It needs to be said that these data relate to microsatellites and how many loci. AR needs to be defined. NA is presumably mean number of alleles per locus?

Experimental design

No comments

Validity of the findings

Discussion of molecular dating in the Introduction should indicate that the premises on which the dating was carried out may be incorrect. See the recent consideration of this point by Herman & Searle (2011; Proc R Soc B278, 3601) and references therein.

In the Results (bottom of p 14) – I’m not convinced that there is either non-random mating or biased introgression within the hybrid zone. Most samples will not be precisely in the center of the hybrid zone, therefore one would still expect a strong association of lineage A nuclear alleles with mitochondrially lineage A individuals, and likewise for lineage B. It’s not appropriate to think of the hybrid zone as some isolated panmictic population. Individuals can still be mating randomly among those individuals that they meet, but the proportions of individuals of the different lineages will change as you go across the hybrid zone.

In the Discussion (bottom of p 18), I’m not convinced that individuals in the center of the hybrid zone form a separate evolving genetic entity from the parental forms. Inevitably there will be greater similarity among the hybrids than between the hybrids and pure races as shown in the BAPS results, but this can easily be explained just through interbreeding of the pure races and a typical tension zone structure (i.e. with hybrid disadvantage: Barton and Hewitt 1985 Ann Rev Ecol Syst 16, 113). I'm not convinced that there is need to invoke emergence of a new hybrid race in this case.

Additional comments

This is large scale study of hybridization in rabbits in Iberia in terms of numbers of individuals, with individuals being typed for mitochondrial lineage and microsatellites. There have been other studies on the hybrid zone between the two major subspecies, but a further analysis is welcome, and a range of statistical methods are employed. While much of the new data and old findings are interpreted reasonably, I take issue with three points as indicated in the previous section. These should be addressed in the revision together with the minor corrections.

---

## Round 0.2 · Minor Revisions

Dear Dr. Alda -

I have gotten feedback on your revised manuscript from one of the previous reviewers and this reviewer feels that the paper is now acceptable. However, the reviewer indicated that the English could still be improved throughout. I would like you to look through the manuscript carefully and have it read by a native English speaker. Once you submit a revised manuscript, I will likely accept it for publication.

Thanks again for your patience while I solicited this second review.

Scott

Reviewer 1 ·

Basic reporting

No Comments

Experimental design

No Comments

Validity of the findings

No Comments

Additional comments

No Comments

---

## Round 0.3 · accepted · Accept

This manuscript is now ready to go and I think offers a useful contribution to the literature on speciation in rabbits and on the Iberian peninsula. Thank you for having the English corrected and for acknowledging A. Peterson.